# Comparison of Fracture Toughness in the Coarse-Grain Heat-Affected Zone of X80 Pipelines Girth-Welded under Conventional and Ultra-Low Heat Input

**DOI:** 10.3390/ma15217701

**Published:** 2022-11-02

**Authors:** Shuo Liu, Lingzhi Ba, Chengning Li, Xinjie Di

**Affiliations:** 1School of Materials Science and Engineering, Tianjin University, Tianjin 300350, China; 2Central Research Institute, Baoshan Iron & Steel Co., Ltd., Shanghai 201900, China; 3Tianjin Key Laboratory of Advanced Joining Technology, Tianjin University, Tianjin 300350, China

**Keywords:** X80 pipeline steel, girth welded, heat input, coarse-grained heat-affected zone, fracture behavior

## Abstract

The coarse-grain heat-affected zones (CGHAZs) of X80 girth-welded steel pipelines are prone to embrittlement, which has an extremely adverse effect on their structural integrity. In the present work, the fracture behavior of the CGHAZs of X80 girth welds under the conditions of conventional and ultra-low heat input was studied. The fracture toughness of CGHAZs was evaluated using the crack tip opening displacement (CTOD) test at −10 °C, and the fracture behavior mechanism of CGHAZs were clarified by analyzing microstructural characteristics at prefabricated fatigue cracks containing fracture cloud image, scanning electron microscopy (SEM), and electron back-scatter diffraction (EBSD) figures. The results illustrate that the average fracture toughness (CTOD) value of the ultra-low heat input CGHAZ is 0.6 mm, and the dispersion of CTOD values is small, while the CTOD value of conventional heat input is only 0.04 mm. The ultra-low heat input makes the high-temperature residence time of the coarse-grained region short, reduces the proportion of prior austenite grain boundaries, and inhibits the formation of strip-like bainite and island-like M-A components. The reduction of these deleterious ductile microstructures increases the plastic reserve and deformation capacity of the CGHAZ.

## 1. Introduction

For pipeline applications in oil and gas transportation, high strength, high toughness, long service life, and diversity are the trends in the development and innovation of pipeline steel materials [1]. Utilization of high-grade pipe steels with excellent combined mechanical properties is imperative to the economy and environment because the total tonnage of steel structures and thus the carbon dioxide (CO_2_) emissions associated with steel manufacture and transport can be significantly reduced. The accuracy of predicting a material’s fracture toughness plays an important role in any assessment of structural integrity linked to design procedure validation or critical defect analysis. Such structural assessments are based on the assumption that the fracture resistance of a material of interest is equivalent to the resistance due to specimen toughness [2,3].

Girth welding is a major and key process of connecting high steel grade line pipes, and its quality, efficiency, and cost determine the comprehensive cost and safety in operation. In recent years, automatic gas metal arc welding (GMAW) has become one of the main welding methods in the pipeline construction field. As far as multi-pass welding is concerned, the former welding pass will be influenced by the following pass, which complicates the analysis of the heat affected zones (HAZs) for the girth weld [4]. Khalaj et. al. successfully predicted austenite grain size and toughness in an Nb/Ti micro-alloyed pipeline steel weld heat affected zone using the grain growth model and artificial neural network model, respectively [5,6]. In this study, combined with the API Specification 5 L, the weakest area of coarse-grained HAZ (CGHAZ), i.e., fusion line (FL) +0.5 mm, is deeply investigated. In terms of the comprehensive evaluation of the girth weld joint, the fracture toughness is the imperative safety index in the processes of line pipe construction and operation [7].

The fracture toughness can not only characterize the potential crack initiation and stable propagation resistance, but can also serve as the key input for engineering critical assessment (ECA), aiming at the assessment of the maximum defect tolerance for the obtained girth weld joints. The indicators, including critical crack tip stress intensity factor (K_Ic_), critical J integral (J_Ic_), and crack tip opening displacement (CTOD, δ_Ic_), are frequently relied on to evaluate the fracture toughness of structural materials and their weld joints. However, as for the line pipe welded structures, CTOD is generally used to characterize fracture toughness [8,9,10].

Even if recently some achievements in the fields of welding quality and girth weld performance improvement regarding X80 pipe construction have been made, there still exist the problems of the fluctuation of the CTOD values, such as the existence of some quite low individual CTOD values. It is necessary to carry out an in-depth investigation of the fracture behavior of the weakest area for the girth weld joint, combining materials themselves with welding procedures, for better understanding the optimal direction of the line pipe construction [11,12,13].

In general, the heat input range for automatic GMAW is 0.5~1.0 kJ/mm in the welding process of long-distance oil and gas line pipes [14]. However, for the pipeline supplier, field girth weldability of the pipeline is required by many abroad consumers, in which less than 0.25 kJ/mm ultra-low heat input is specified with the purpose of satisfying the whole possible heat input range in the construction fields, especially for root-girth welding. The CTOD of CGHAZ needs to be analyzed systematically in order to understand the influence of the ultra-low heat input on fracture toughness of girth welds by automatic welding [14,15,16].

In this study, two typical heat input ranges including <0.25 kJ/mm and 0.5~1.0 kJ/mm were selected for the girth welding of X80 pipe steel, and the CTOD values of CGHAZ for welded pipes were investigated, aiming at the safety assessment for the pipe engineering projects. The CGHAZ microstructure and fracture surfaces located at the tip of the pre-fabricated fatigue crack were studied by scanning electron microscope (SEM) and electron back-scattered diffraction (EBSD). What is more, the fracture behaviors during the CTOD test were discussed in detail and the relationships between microstructural characteristics and toughness were studied based on the EBSD analysis and fracture contour map.

## 2. Materials and Methods

The X80 pipeline steel used in this study was a longitudinal submerged arc welding pipe with an outer diameter of 1422 mm and a thickness of 21.4 mm; its chemical composition is shown in Table 1. The girth welding was completed through all-position automatic welding by the welding system from Lincoln Electric, which constituted a Power Wave S500 GMAW power source, a Helix M85 welding head, and APEX 3000 remote controller. The groove of the girth weld was a composite-V type, as described in Figure 1. The welding materials (consumables) used in the girth welding can be seen in Table 2. The ER70S-G and ER80S-G were selected for the root pass and hot/fill/cap passes, respectively. The ultra-low heat input (<0.25 kJ/mm) and conventional heat input (~0.7 kJ/mm) were designed for comparison. In the practical welding (Table 3), (80%Ar + 20%CO_2_) shielding gas with the gas flow rate of 25~40 L/min were used. After welding, the radiographic testing (RT) was used to confirm no internal defects.

The macro-morphology and microstructure located at the CGHAZs after girth welding with different heat inputs are shown in Figure 2. It can be observed that the weld metal is well connected, and there was no unfused metal, slag inclusion, or other defects, as shown in Figure 2a,b. The microstructure types of conventional and ultra-low heat input CGHAZs are similar, there are a large number of prior austenite grain boundaries, and the lath-like bainite and ferrite mixed structures of different directions are distributed inside the prior austenite grains. In addition, with the increase of heat input, the size of prior austenite increased from 30 µm to 64 µm, and the size of the internal distribution of lath bundles and martensite-austenite components (M-A) increased greatly.

The girth weld samples were prepared for the CTOD tests based on the single-edge notch bending (SENB) experiments, and the notch in transverse was used. The size of the sample for the CTOD test was B × W (B: wall thickness, B = W = 18.5 mm). The location of the prepared notch in the sample is displayed in Figure 3. Note that the notch has to be positioned through the etched sample, and the tip of the pre-fabricated fatigue crack should be located within 0.5 mm of the fusion line to guarantee the validity of the CTOD test. The standard used for the whole experiment is ISO 15653.

The CGHAZ microstructure and fracture surfaces located at the tip of the pre-fabricated fatigue crack were studied by SEM and EBSD. EBSD specimens were mechanically polished and electrolytically etched in 5 vol% perchloric acid and 95 vol% ethyl alcohol at 20 °C and 30 V. The post treatment of the scan data was carried out in orientation imaging microscopy (OIM) software. Using the super depth of field microscope capture and ZEISS smart zoom 5 software, the topography located at the front section of the fracture and the contour map was obtained.

## 3. Results and Discussion

### 3.1. Fracture Toughness (CTOD)

Table 4 lists the CTOD results of the CGHAZs under two different welding heat inputs at −10 °C. From the results, it is evident that the CTOD values of the CGHAZ under ultra-low heat input, which is the ductile fracture, were higher. The ultra-low heat input can obtain better fracture toughness compared to the CTOD values under conventional heat input, which is the brittle fracture.

After inspection for the front microstructure of pre-fabricated fatigue cracks, it was found that all the fatigue cracks expanded in the areas of the CGHAZs, which proves the validity of the experiments. From the optical microscope image of the crack propagation path in Figure 4, it can be seen that the crack propagation path of ultra-low heat input is very tortuous, while the propagation path of conventional heat input is almost a straight line until the unstable fracture. This shows that the ultra-low heat input CGHAZ microstructure undergoes significant plastic deformation during crack propagation, which has a strong hindering effect on cracks, thereby obtaining excellent fracture toughness values [17]. The conventional heat input is just the opposite, and it breaks instantaneously after reaching the maximum force, resulting in a very low CTOD value.

### 3.2. Microstructural Analysis

In order to clarify the possible reasons for the fracture behavior mechanism during the CTOD test with different heat inputs, the microstructures of prefabricated fatigue crack tips using two heat inputs were observed by scanning electron microscopy (SEM) technology (Figure 5). Two types of bainite were formed under ultra-low heat input [18]: (1) strip-like lath bainite, where M-A particles are dispersed along the grain boundaries, (2) the mixed microstructure of small blocky bainite and ferrite, where martensite-austenite (M-A) islands are distributed in the grains. In the microstructure, no obvious prior austenite grain boundaries appeared. Meanwhile, the pronounced plastic deformation could be observed at the fatigue crack tip because the strip-like lath bainite absorbs partial energy in the process of crack propagation, which is beneficial to blunting the crack tips to some degree [19]. Therefore, the strip-like lath bainite can resist the further propagation of the crack, and improve the CTOD value accordingly.

Under conventional heat input conditions, relatively coarse prior austenite grain boundaries (PAGBs) were generated, and the bainitic laths were distributed in different directions in the prior austenite grains, together with long and parallel M-A constituents. The coarse bainite located in the CGHAZ area had a large crystallographic effective size, which increased the linear propagation path of the cleavage cracks, and undoubtedly impaired the fracture toughness [20,21]. At the same time, no plastic deformation occurred at the fatigue crack tips, and the major crack extended along the coarse PAGBs [22]. Furthermore, secondary cracks were found at the board of bainite matrix and M-A constituents, which seriously deteriorated the fracture toughness as well.

### 3.3. EBSD Analysis

Figure 6 displays the microstructures in the front of prefabricated fatigue cracks under two typical heat inputs by EBSD analysis. From the orientation maps or inverse pole figure (IPF) maps, seen in Figure 6a,d, it is evident that the microstructure of the CGHAZ region under both heat inputs had an irregular crystallographic orientation with a random distribution. The distribution of grain sizes for two CGHAZs can be seen in Figure 7a,d. It can be seen that the ultra-low heat input CGHAZ structure was very fine, and the grain size was much lower than that of the conventional heat input.

In terms of the grain boundaries, the high angle grain boundaries (HAGBs) (>15°) and low angle grain boundaries (LAGBs) (2~15°) are represented by the blue and red lines in Figure 6b,e, respectively. Figure 7b,e present the histograms of grain boundary angle distributions for ultra-low and conventional heat input, respectively. The distribution fraction of large and small angle grain boundaries is similar between the two heat inputs. However, the small angle grain boundaries of the conventional heat input CGHAZ were mainly the grain boundaries of the M-A components distributed on the bainite and the matrix, which exist in the organization itself, while the low-angle grain boundaries of the ultra-low heat input CGHAZ were formed by dislocation entanglement caused by plastic deformation during the CTOD test [23]. The ratio of large and small angle grain boundaries has a strong influence on the properties of welds and heat-affected zones. Moshtaghi et. al. proposed the high HAGB density results in a lower H diffusion coefficient and higher density of relatively strong HAGB traps [24]. Di et al. suggested that increasing the proportion of HAGB can significantly increase the strength and toughness of the material at the same time [25]. The HAGB ratio of ultra-low heat input is higher than that of the conventional heat input, which can also be demonstrated through the hardness values of these heat inputs: the average hardness of CGHAZ with ultra-low heat input is 294 HV, while that of the normal heat input is 267 HV. The hardness is related to the structure type and the ratio of LAGB and HAGB, and the microstructure types of the two heat inputs are consistent, so the ultra-low heat input has more HAGB, resulting in higher hardness values.

Furthermore, Figure 6c,f illustrate the distribution of kernel average misorientation (KAM) maps in the CGHAZs under the two heat inputs. The KAM diagram can intuitively represent the deformation degree of the prefabricated fatigue crack tip. The greater the deformation degree, the higher the KAM value. The KAM value of the ultra-low heat input CGHAZ is larger as it is closer to the crack, while the KAM distribution map of the conventional heat input CGHAZ is almost all blue, and the distribution is very uniform. Figure 7c,f are histograms of KAM numerical distributions for two heat inputs. It can be found that the average KAM value of ultra-low heat input is 1.12, far exceeding the average KAM value of conventional heat input, which is 0.68.

It can be understood that the crack front on the left side of the CGHAZ under ultra-low heat input has undergone an observable deformation, as shown in Figure 6. With the redistribution and aggregation of dislocations during plastic deformation, the average orientation difference in this region is apparent, and the degree of stress concentration is high during the experimental loading process [26].

However, in the adjacent regions where no plastic deformation occurs, there was no significant local orientation difference. From the grain size distribution (Figure 7), because it is located in the CGHAZ, there were a number of large-sized grains at the front of the crack, but the proportion of small-sized grains was also high (Figure 4). At the same time, in the process of blunting and plastic deformation at the front of the crack, the glide and pile-up of dislocation were generated, and the tiny secondary unit cells were formed inside the original grain, resulting in a large number of small-angle grain boundaries [27]. Given the small overall grain size, a large number of low-angle grain boundaries also did not cause severe embrittlement of CGHAZ.

In summary, under the condition of ultra-low heat input, the small orientation difference between the bainite matrix and the M-A constituent in the CGHAZ and the obtained finer grain size (Figure 7) enable this important area to control plastic deformation and fracture resistance well [25]. In contrast, the conventional welding heat inputted CGHAZ did not experience evident blunting and plastic deformation during the propagation of the prefabricated fatigue crack, and the crack propagated along the path with the lowest energy consumption after loading, which was consistent with the SEM result (Figure 4 and Figure 5). Coarse lath bainite without plastic deformation and a large number of LAGBs inside the grains cannot effectively prevent crack propagation [24].

There is a local orientation difference between the PAG boundary (Figure 5), the lath bainite matrix, and the strip-like M-A constituent, and the stress concentration occurred at the boundary of the M-A constituent, which is a potential location for secondary crack initiation [19], and even promotes the crack propagation, reducing the plastic deformation capacity and fracture resistance at this location. From the perspective of grain size and grain boundary distribution, the grains in the CGHAZ under conventional heat input were large and nonuniform, and the proportion of HAG boundaries was extremely low, which is not conducive to hindering crack propagation.

### 3.4. Fracture Behavior Analysis

In the CTOD experiment, the three-point bending test was executed on prefabricated fatigue cracks in order to investigate the plastic deformation behavior of the prefabricated fatigue crack, and characterize the ability of the studied materials or their HAZs to control the crack propagation and crack arrest [28]. Therefore, a contour map at the front of the prefabricated fatigue crack can clarify its performance in elastic-plastic deformation, and indirectly stand for the CTOD fracture toughness, as shown in Figure 8.

Figure 9 shows the fracture contour map and fracture morphology of the prefabricated fatigue crack front of the CGHAZs under two typical heat inputs. Table 5 shows the fracture contour map at the front of the prefabricated fatigue crack of the CGHAZs under two typical heat inputs. The detailed values are calculated based on the fracture morphology. The larger fracture contour map is closely related to the elastic-plastic deformation ability and initial blunting of the front of the prefabricated fatigue crack and during the process of crack stable expansion, representing the CTOD performance [29]. From the fracture surface at the prefabricated fatigue crack tip (Figure 9c,d), the morphology of both heat inputs consisted of a dimple shape. However, the size and depth of the dimples vary widely, the ultra-low heat input specimens having large-sized, large-depth dimples, filled with small-sized dimples around the large dimples. The conventional heat input was also composed of dimples, but the dimples were small and shallow, and the energy value offset in the crack propagation was less, so the fracture toughness is low.

Combining the CTOD results listed in Table 4, it can be seen that there is a clear positive correlation between the CTOD fracture toughness and the fracture contour map at the front the prefabricated fatigue crack, especially in high brittle materials and their structures with low CTOD performance. Excellent fracture toughness must be accompanied by larger plastic deformation [30], which is reflected in the macroscopic height cloud map. For a given sample size, the greater the height difference at the crack front, the better the fracture toughness and the higher the CTOD value. The height difference of ultra-low heat input was more than 3 mm, and the corresponding CTOD average value reaches 0.6 mm (Table 4). The testing results show that when the CTOD values of a material or its structure exceeds 0.254 mm and turns to the stage of complete plastic fracture, the correlation between CTOD value and fracture height difference is supposed to be impaired.

## 4. Conclusions

The girth welds under two typical heat inputs, i.e., ultra-low heat input and conventional heat input, were designed for this study in order to clarify the influence of the heat input on the fracture toughness of most the sensitive CGHAZ of two welds.

(1) Two heat input ranges including <0.25 kJ/mm and 0.5~1.0 kJ/mm in girth welding can both be completed with high quality.

(2) The CGHAZ under ultra-low heat input has a higher and more stable CTOD performance, which excludes the possibility of quenching embrittlement, which leads to reduced fracture toughness for the CGHAZ of the girth weld.

(3) The coarse prior austenite grain boundary at the CGHAZ under ultra-low heat input was reduced. The fine lath bainite phases and the strip-like M-A constituents distributed along the grain boundaries improved the plastic reserve and deformation capability of the sensitive CGHAZ, effectively leading to increased fracture toughness.

(4) The larger height difference at the front of the prefabricated fatigue cracks and the orientation difference of the microstructure can be applied to reflect the fracture toughness value.

## Figures and Tables

**Figure 1 materials-15-07701-f001:**
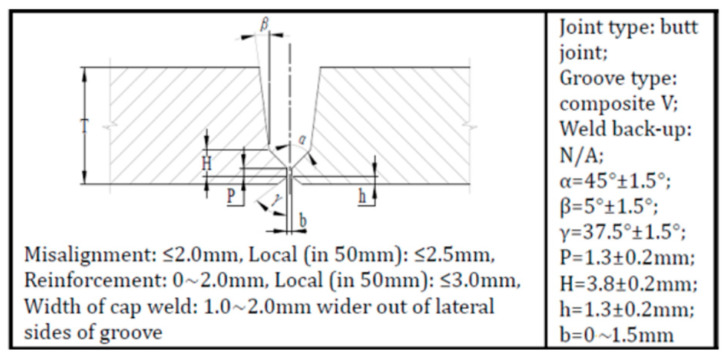
Design of composite V-groove.

**Figure 2 materials-15-07701-f002:**
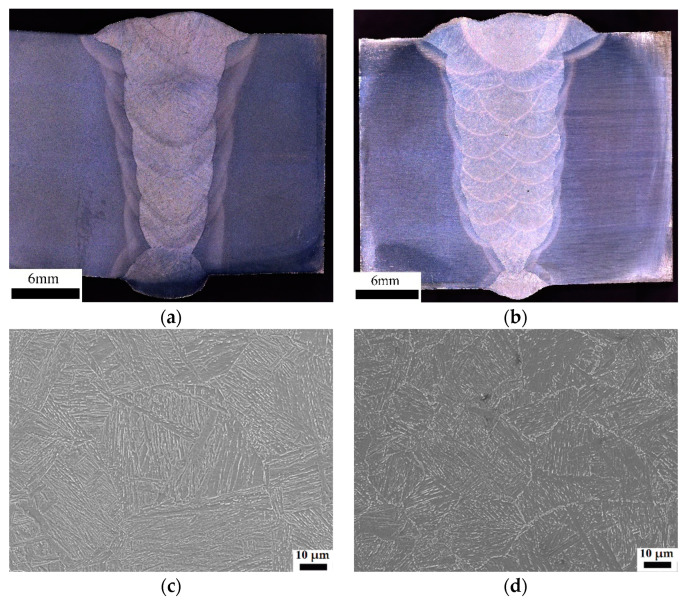
Macro-morphology of girth welded joints and microstructure of CGHAZs with different heat inputs: (**a**,**c**) conventional heat input; (**b**,**d**) ultra-low heat input.

**Figure 3 materials-15-07701-f003:**
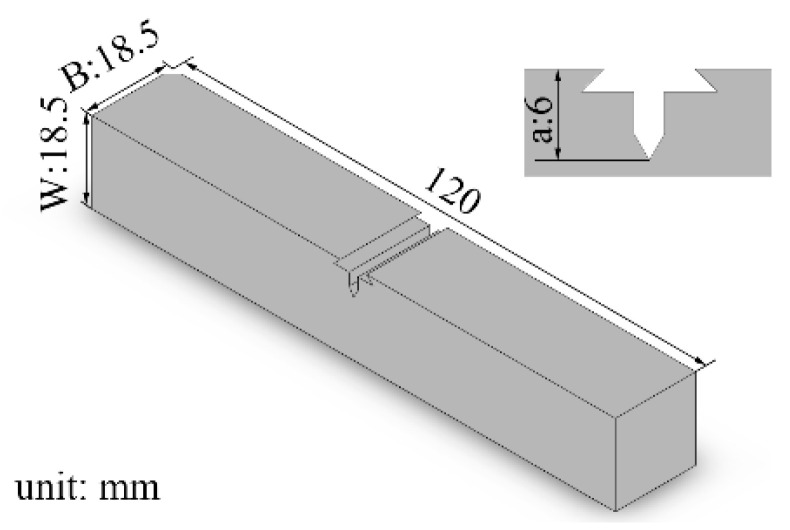
Description of prepared notch form of CTOD fracture toughness specimen (W: wall thickness of the steel pipe; a: depth of prepared notch).

**Figure 4 materials-15-07701-f004:**
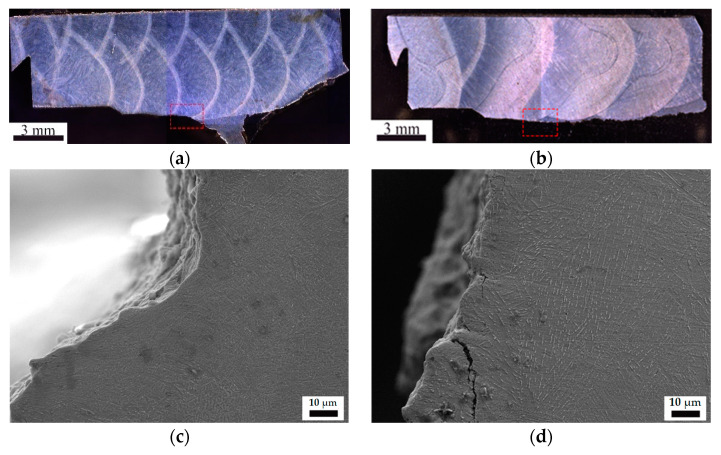
Optical microscope image of crack propagation path and SEM microstructure of plastic crack initiation position in CTOD test. (**a**,**c**) Ultra-low heat input; (**b**,**d**) conventional heat input.

**Figure 5 materials-15-07701-f005:**
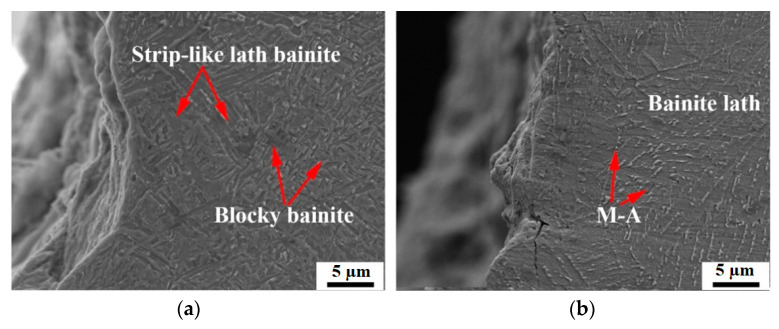
SEM microstructure at prefabricated fatigue crack tips using two heat inputs: (**a**) ultra-low heat input; (**b**) conventional heat input.

**Figure 6 materials-15-07701-f006:**
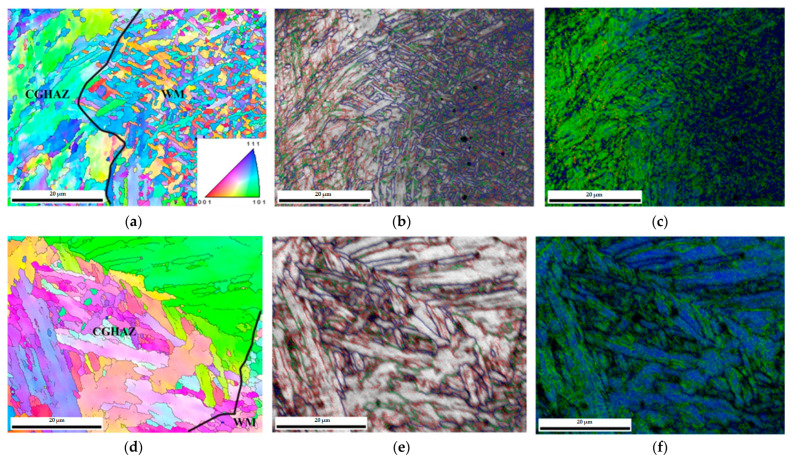
EBSD analysis of pre-fabricated crack tips under two heat inputs: (**a**–**c**) IPF map, size-angle grain boundary distribution map and KAM map for ultra-low heat input; (**d**–**f**) IPF map, size-angle grain boundary distribution map and KAM map for conventional heat input.

**Figure 7 materials-15-07701-f007:**
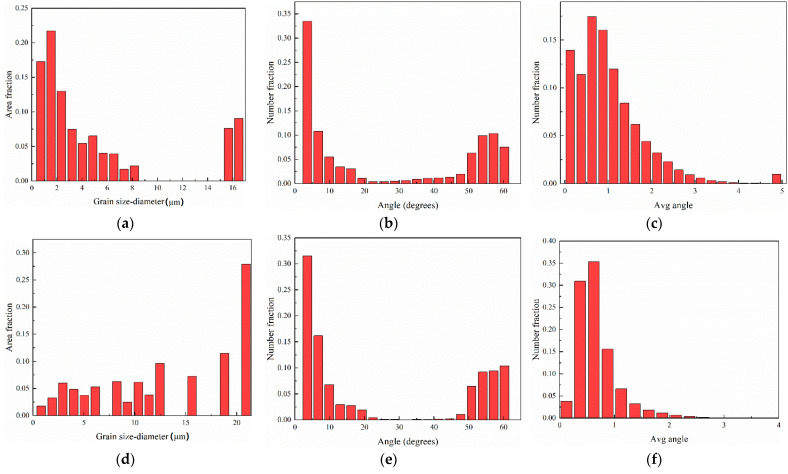
(**a**,**d**) Grain size, (**b**,**e**) size-angle grain boundary and (**c**,**f**) KAM distribution for ultra-low and conventional heat input, respectively.

**Figure 8 materials-15-07701-f008:**
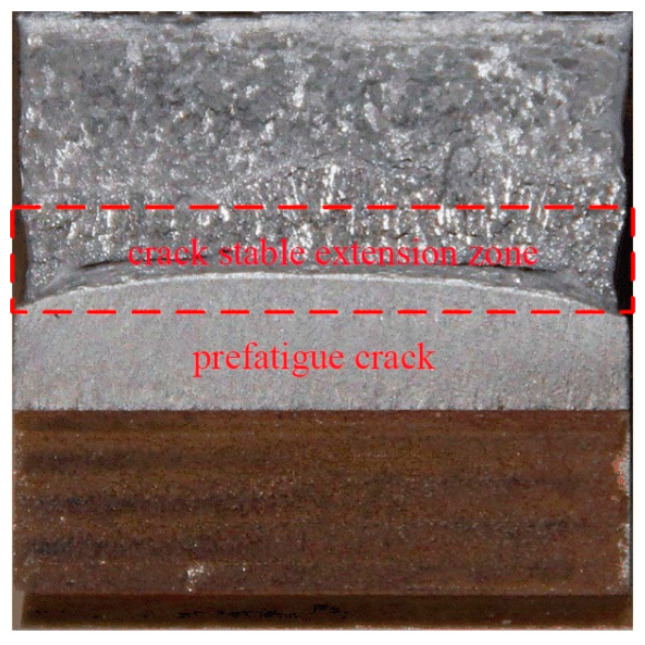
Location of the cloud image analysis of the fracture front of the prefabricated fatigue crack.

**Figure 9 materials-15-07701-f009:**
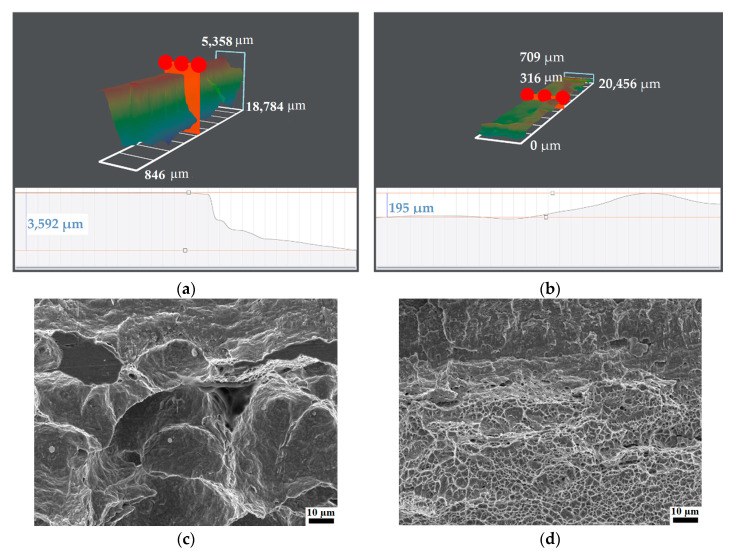
Contour map and fracture morphology of prefabricated fatigue crack tip under two heat input: (**a**,**c**) ultra-low heat input; (**b**,**d**) conventional heat input.

**Table 1 materials-15-07701-t001:** Chemical composition and carbon equivalent of target X80 pipe steel (wt.%).

C	Si	Mn	P	S	Cr	Ni	Nb	Ti	CE_IIW_	P_cm_
0.05	0.2	1.7	<0.01	<0.003	0.22	0.18	0.06	0.02	0.44	0.17

**Table 2 materials-15-07701-t002:** Welding materials used in girth welding.

Root Pass		Hot/Filler/Cap Pass	
Specification	Type	Specification	Type
AWS A5.18 ER70S-G	Bohler SG3-P	AWS A5.18 ER80S-G	Bohler SG8-P

**Table 3 materials-15-07701-t003:** Two different heat input automatic circumferential seam welding process parameters.

Sample No.	Pass	Weld Direction	Current (A)	Voltage (V)	Welding Speed (mm/min)	Heat Input (kJ/mm)
Ultra-low heat input	Root	Downward	180–200	20–22	950	0.25
	Hot		175–250	18–21	800–1200	<0.25
	Filler		175–250	18–21	800–1200	<0.25
	Cap		175–250	18–21	800–1200	<0.25
Conventionalheat input	Root	Downward	180–200	19–21	700	0.33
	Hot		160–240	22–26	380–510	0.65
	Filler		150–240	22–26	330–460	0.71
	Cap		150–210	20–25	410–600	0.50

**Table 4 materials-15-07701-t004:** CTOD results of CGHAZs under two welding heat inputs.

	Ultra-Low Heat Input			Conventional Heat Input		
Sample No.	U-1#	U-2#	U-3#	N-1#	N-2#	N-3#
CTOD (δ, mm)	0.61	0.61	0.58	0.06	0.01	0.03
Valid/invalid	Valid	Valid	Valid	Valid	Valid	Valid

**Table 5 materials-15-07701-t005:** Fracture height difference of prefabricated fatigue cracks under two typical heat inputs.

Welding No.	Height Difference (μm)	
	CGHAZ-1	CGHAZ-2
Ultra-low heat input	3109	3592
Conventional heat input	475	195

## Data Availability

Not applicable.

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
