# Peer review of "Comparison of Fracture Toughness in the Coarse-Grain Heat-Affected Zone of X80 Pipelines Girth-Welded under Conventional and Ultra-Low Heat Input"

_materials, 2022, doi:10.3390/ma15217701_

Round 1

Reviewer 1 Report

The presented manuscript seems to be interesting for readers of the Materials journal, it is written in a good manner and suits the requirements of the journal. It can be accepted for publication after minor corrections listed below.

- The authors have stated in line 116 of the text that:”The size of the sample for the CTOD test was B×W (B: wall thickness, B=W=10mm); While according to Figure 4, the width of the sample is more than 10 mm.

- It is suggested to display the initial dimensions of the CTOD sample (W: Wall thickness of the steel pipe; a: Depth of prepared notch, B: wall thickness) in Figure 3. Also, the three-dimensional shape and schematic of the notched sample should be displayed.

- The location of crack growth and separation surfaces should be investigated by characterization methods.

- From lines 216 to 246, the authors have given results that require more evidence and confirmations to prove.

- The software and method used to determine and draw the contour map in Figure 9 should be given in detail in the text.

- The authors stated in conclusion 4 that:” The larger height difference at the front of the prefabricated fatigue cracks and the orientation difference of the microstructure can be applied to reflect the fracture toughness value.” This result needs to provide more evidence and reasons to prove it.

Literature review is not sufficient and authors must review and cite more papers in the field of phase transformations in HAZ and relationship between structure and properties in pipeline steels and especially newly published ones. Doing this, reviewing and citing the following refs could be helpful: Materials Science and Technology, 30(4), 2014, 424-33, Journal of Mining and Metallurgy, Section B: Metallurgy, 51, 2015, 173-178.

Reviewer 2 Report

In the study, the comparison of fracture toughness of CGHAZ of X80 pipeline 2 and its effect under ultra-low heat input were investigated.

When the results were examined, the average fracture toughness value was determined at the ultra-low heat input and it was seen that the distribution was small and the conventional heat input was measured. Here, it is discussed to increase the plastic reserve and deformation capacity by reducing harmful ductile microstructures.

The number of references is not enough. Other studies with a high self-citation rate should also be included.

Reviewer 3 Report

The manuscript refers to the comparison of fracture toughness in CGHAZ of X80 pipeline girth welded under conventional and ultra-low heat input. The following corrections are required before publication:

1. CGHAZ in the title is not a proper word. It is recommended that the title be reworded.

2. Scale bars for Figs. 4c and d should be added 

3. Some EBSD images should represent the crack path.

4. Hardness tests performed at different locations of the specimen can confirm the results.

5. During welding, the types of grain boundaries may change. The effect of the grain boundary change on the properties should be explained in detail in the following paper. https://doi.org/10.1016/j.ijhydene.2022.04.260

Round 2

Reviewer 1 Report

As authors have performed an adequate revise, the manuscript might be accepted for publication in the journal of Materials

Reviewer 3 Report

The manuscript can be published in this format.